# Self-Supervised Fair Representation Learning without Demographics

**Junyi Chai, Xiaoqian Wang**[*]
Elmore Family School of Electrical and Computer Engineering
Purdue University
West Lafayette, IN 47906
{chai28,joywang}@purdue.edu

## Abstract

Fairness has become an important topic in machine learning. Generally, most literature on fairness assumes that the sensitive information, such as gender or race, is present in the training set, and uses this information to mitigate bias. However, due to practical concerns like privacy and regulation, applications of these methods are restricted. Also, although much of the literature studies supervised learning, in many real-world scenarios, we want to utilize the large unlabelled dataset to improve the model's accuracy. *Can we improve fair classification without sensitive information and without labels?* To tackle the problem, in this paper, we propose a novel reweighing-based contrastive learning method. The goal of our method is to learn a generally fair representation without observing sensitive attributes. Our method assigns weights to training samples per iteration based on their gradient directions relative to the validation samples such that the average top-$k$ validation loss is minimized. Compared with past fairness methods without demographics, our method is built on fully unsupervised training data and requires only a small labelled validation set. We provide rigorous theoretical proof of the convergence of our model. Experimental results show that our proposed method achieves better or comparable performance than state-of-the-art methods on three datasets in terms of accuracy and several fairness metrics.

## 1   Introduction

As machine learning systems are increasingly used for automated decision making with social impact, discrimination across different demographic groups has become an important concern (Ntoutsi et al., 2020). There have been growing evidence of disparities across different demographic groups in real-world machine learning systems. Research on the COMPAS dataset shows a strong correlation between recidivism prediction and race, where black individuals are more likely to be classified as high risk (Larson et al., 2016). Gender bias has been identified in image captioning models (Hendricks et al., 2018), where the model is likely to recognize the person in the image (without considering the person's appearance) as a man if the scene shows snowboarding. It is crucial to ensure that algorithm makes fair predictions without explicit or implicit consideration of biased information, so as not to reveal or amplify social biases.

Due to distributional disparity, historical bias, or indirect discrimination (Hajian and Domingo-Ferrer, 2012), simply removing sensitive attribute during training is not enough to guarantee fairness. In response, methods (Hardt et al., 2016; Creager et al., 2019; Bahng et al., 2020) have been proposed to achieve fairness under certain fairness notions. Generally, these methods assume one predefined sensitive attribute, and apply fairness constraints based on such information. However,

---

[*]Corresponding author.

36th Conference on Neural Information Processing Systems (NeurIPS 2022).

in real-world scenarios, due to privacy or legal concern, it might be infeasible to collect or use the sensitive information. For example, General Data Protection Regulation (GDPR) imposes heightened prerequisites to collect and use protected features (Voigt and Von dem Bussche, 2017). Besides, for many automatic decision-making systems, it is important to make sure that the system is not biased against several sensitive attributes. For example, on the COMPAS dataset, we may expect the classifier to be fair with respect to both race and gender. Under such scenarios, conventional methods on fairness that are formulated with one predefined sensitive attribute and one predefined fairness metric (Hardt et al., 2016; Krasanakis et al., 2018) would fail to work, and it is important to study the problem of fairness without demographics.

Current approaches on fairness without demographics (Hashimoto et al., 2018; Lahoti et al., 2020) have been following the idea of Max-Min problem. These works are proposed under a fully supervised setting. In many real-world scenarios, however, we are often given an unlabelled dataset, or only a small portion of labelled samples are available, and to improve model generalization, we always want to utilize the large unlabelled set during training. Thus we ask: *Can we improve fair classification without sensitive information and without labels?*

One straightforward solution to the problem is contrastive learning. Without using labels during training, contrastive learning aims to learn a compact representation in $\mathbb{R}^d$, which minimizes the distance between positive pairs and maximizes the distance between negative pairs based on the similarity between data. However, since we can only access data without labels in contrastive learning, we would inevitably sample false negative pairs. Instead, we consider the following relaxed problem: *Given a small set of labelled samples and a large unlabelled set, can we improve fair classification without sensitive information using contrastive learning?*

In this paper, we propose a fair contrastive learning method without sensitive attributes. Compared with previous works on fairness without demographics which are fully supervised, our method follows the Min-Max fairness constraints and only requires a small set with labels. During training, we calculate the gradients of training samples at each iteration and compare them with the average gradient of validation samples with top-$k$ validation loss. If one sample has a positive contribution to the decrease of validation loss, we would apply higher weight on the sample, and otherwise we would discard the training sample at the current iteration. In this way, we aim to approach an optimal distribution of training samples that minimizes the top-$k$ validation loss.

We summarize our contribution as follows:

1. We formulate a novel contrastive learning method with gradient-based reweighing to learn fair representations without demographics. We also theoretically prove the convergence of our method.

2. Our model effectively incorporates limited supervised information into self-supervised training and achieves comparable or better classification results than fully supervised methods.

3. We validate the effectiveness of our method in improving fairness on three benchmark datasets under various sensitive attributes.

## 2 Related Work

**Fairness in Machine Learning**: Fairness has been widely studied in machine learning area. Generally, fairness methods can be divided into two categories: individual fairness (Mukherjee et al., 2020; Ilvento, 2019; Sharifi-Malvajerdi et al., 2019; Petersen et al., 2021) and group fairness (Feldman et al., 2015; Hardt et al., 2016). Our method follows the idea of group fairness. Different fairness notions including disparate impact (Kamiran and Calders, 2012) and equalized odds (Hardt et al., 2016) have been proposed to quantify disparities of machine learning models. Kleinberg et al. (2016) argue that disparities in machine learning arises due to biased data, instead of biased classifiers. In response, different methods have been proposed to modify input features and eliminate potential bias in downstream tasks. Creager et al. (2019) propose to quantify the correlation between sensitive information and input features using KL-divergence and modify the input feature using VAE. Sattigeri et al. (2018) and Jang et al. (2021) propose to generate a fair dataset using generative models. Calmon et al. (2017) learn a mapping from training distribution to an underlying fair distribution and relax the problem to a convex optimization problem. Jiang and Nachum (2020) adjust training labels based on the desired specific fairness constraints. Other than preprocessing, methods on fairness fall in two categories: inprocessing and postprocessing. Zafar et al. (2017) propose to reduce the

fairness constraints to linear correlation constraints, and use such regularization term during training. Kim et al. (2019) learn a fair classifier through adversarial training with an extra discriminator. Adel et al. (2019) propose to add one extra layer on original network and use adversarial training to achieve fairness. Chai and Wang (2022) propose to learn adaptive weights to balance among different demographic groups. As for postprocessing, the methods adjust the decision threshold of a classifier based on the expected fairness constraints Hardt et al. (2016); Jang et al. (2022). All these methods require sensitive information to formulate fairness metrics. A more related problem is fair classification under noisy sensitive attribute (Gupta et al., 2018; Lamy et al., 2019; Wang et al., 2020), where the goal is to improve fairness on clean data under partially flipped training sensitive attributes.

**Fairness without Demographics**: Fairness without demographics has been less studied. Some works solve this problem by using proxy sensitive features. Grari et al. (2021) propose to approximate the sensitive attributes with VAE. Yan et al. (2020) propose to use clustering information as an approximation of sensitive information. However, such methods either require the strong assumption that the clustering information is correlated with the sensitive information, or are prone to estimation bias. Besides, such methods cannot generalize to the setting where we expect the classifier to be unbiased w.r.t. several sensitive attributes. Instead, Hashimoto et al. (2018) consider the worst-case distribution over a certain group size and propose to achieve fairness through distributionally robust optimization. Lahoti et al. (2020) propose to reweigh training samples adversarially based on training loss to maximize the average loss. Similar methods have also been discussed in debiased learning (Nam et al., 2020; Liu et al., 2021) which employs a biased model to explore and rectify spurious correlations in training set. Our method follows a similar idea of Min-Max fairness constraints. However, in contrast to the fully supervised setting, our method requires only a small labelled validation set.

**Contrastive Learning**: Learning representations without supervision is a long-standing task, and among these methods, contrastive learning has become increasingly popular. The main idea of contrastive learning is to learn a compact representation of input features on hypersphere such that samples of same labels are as close to each other as possible, while samples of different labels are as far apart as possible. In this way, the trained contrastive encoder can be readily applied to downstream classification tasks by only fine-tuning an extra linear layer on a small labelled set. van den Oord et al. (2018) propose to learn a representation of input features using similarity between different samples and between augmentations of same sample. Chen et al. (2020) show that a strong augmentation is crucial to learn a good representation and prevent the network from exploiting naive cues. Chuang et al. (2020) propose to modify the self-supervised contrastive loss to mitigate the influence of false negative pairs. Xiao et al. (2020) propose to construct separated embedding spaces, such that each embedding space is only invariant to certain augmentations. Verma et al. (2021) propose to use mix-up noise for domain-agnostic augmentation. Tian et al. (2020) prove that optimal augmentations for contrastive learning should be task-dependent and proposes a semi-supervised method to learn effective views for a given task. Apart from self-supervised learning, Khosla et al. (2020) propose two different loss terms to incorporate contrastive loss into a fully supervised learning.

## 3 Self-Supervised Fair Representation Learning without Demographics

### 3.1 Background

Let $\{(\boldsymbol{x}_i, \boldsymbol{y}_i, \boldsymbol{a}_i), 1 \leq i \leq N\}$ be the training set, where for the $i$-th sample, $\boldsymbol{x}_i$ is the input data, $\boldsymbol{y}_i \in \{0,1\}^c$ is the one-hot encoding label, and $\boldsymbol{a}_i \in \{0,1\}^s$ is the sensitive attribute. Let $h(\boldsymbol{x}_i) \in [0,1]^c$ be the output of classifier, then a general fair classification task can be formulated as

$$\arg \min_{h} \frac{1}{N} \sum_{i=1}^{N} \mathcal{L}_{cls}\left(h(\boldsymbol{x}_i), \boldsymbol{y}_i\right), \ \ s.t. \ \ \phi(h) \leq \epsilon,$$

where $\mathcal{L}_{cls}$ is the classification loss and $\phi(h)$ is the fairness constraint. For example, $\phi(h) = \max_{\boldsymbol{a}} |\mathbb{P}(\arg \max_{c'} h(\boldsymbol{x})_{c'} = \text{pos}) - \mathbb{P}(\arg \max_{c'} h(\boldsymbol{x})_{c'} = \text{pos}|\boldsymbol{a})|$ corresponds to fairness constraint of the demographic parity, where $\mathbb{P}(\arg \max_{c'} h(\boldsymbol{x})_{c'} = \text{pos})$ is the probability that $h$ classifies data $\boldsymbol{x}$ to the positive (pos) class. This formulation works when the sensitive information is available in the training set. However, when such information is not available in the training set, a natural question to ask is: how should we formulate the fair classification task without the sensitive information? As

with (Hashimoto et al., 2018) and (Lahoti et al., 2020), we formulate the fairness constraint with sensitive attributes unknown as the following Max-Min problem: maximizing the minimum utility $U$ amongst all sensitive groups. If we choose accuracy as the utility metric, such requirement can be seen as a relaxation of error-based fairness constraints, and the fair classification task without demographics can be formulated as

$$\arg\min_{h} \arg\max_{\boldsymbol{a}'} \frac{1}{|\{i|\boldsymbol{a}_i = \boldsymbol{a}'\}|} \sum_{i \in \{i|\boldsymbol{a}_i = \boldsymbol{a}'\}} \mathcal{L}_{cls}(h(\boldsymbol{x}_i), y_i).$$

And $\boldsymbol{a}'$ can be either surrogate demographic information or estimated with extra assumptions.

## 3.2 Contrastive Learning

Contrastive learning aims at learning a balanced representation on the unit hyper sphere of $\mathbb{R}^d$ based on the similarity of input features. Generally, given a training set of size $N$, during each iteration we sample a mini-batch of size $n$ from the training set and apply random augmentation on each sample twice, resulting in a subset $\{\tilde{\boldsymbol{x}}_i, 1 \leq i \leq 2n\}$. For tabular data, random augmentation includes mix-up and adding Gaussian noise, while for image data, random augmentation includes mix-up, cropping and resizing, cropping, resizing and flipping, color drop, color jitter, rotation, cutout, adding Gaussian noise, Gaussian blur, and Sobel filtering as used in previous work (Chen et al., 2020). For augmentations of same sample, we want the query to match its anchor key over all other samples which are treated as negative. Let $f_\theta$ be a contrastive encoder network with parameter $\theta$ which maps an input onto the unit hyper-sphere of $\mathbb{R}^d$, subsequently used for classification task. For an anchor $\tilde{\boldsymbol{x}}$, a common choice of contrastive loss is InfoNCE (van den Oord et al., 2018):

$$\mathcal{L}_{ctr}(\tilde{\boldsymbol{x}}_i; \theta) = -\log \frac{\exp(\mathrm{sim}(f_\theta(\tilde{\boldsymbol{x}}_i), f_\theta(\tilde{\boldsymbol{x}}_i^{\mathrm{pos}}))/\tau)}{\sum_{j \neq i} \exp\left(\mathrm{sim}\left(f_\theta(\tilde{\boldsymbol{x}}_i), f_\theta(\tilde{\boldsymbol{x}}_j)\right)/\tau\right)} \tag{1}$$

where $\mathrm{sim}$ is the cosine similarity metric, $\tau$ is the temperature scaling hyper-parameter that controls the penalties on hard negative samples, $\boldsymbol{x}^{\mathrm{pos}}$ is the positive sample, and all samples other than $\tilde{\boldsymbol{x}}$ in the mini-batch are negative samples. Ideally, we would like to learn a compact representation of input features such that the embedded features of different labels are pairwise linearly separable. To determine the decision boundary for classification, a common way is to add one extra linear layer after the encoder and train the linear layer on a small validation set with the encoder frozen.

## 3.3 Problem Formulation

We now formulate our problem. Given an unlabelled training set of size $N$ and a small validation set $\{(\boldsymbol{x}_j^{\mathrm{val}}, \boldsymbol{y}_j^{\mathrm{val}}), 1 \leq j \leq M\}$ with $M \ll N$, denote the augmented training set (each sample augmented twice) as $\{\tilde{\boldsymbol{x}}_i, 1 \leq i \leq 2N\}$, our goal is to learn a fair embedding via contrastive learning without observing the sensitive information in either the training or validation set. Inspired by previous work (Hashimoto et al., 2018), we use Min-Max fairness as the surrogate training objective in contrastive learning. To be specific, we propose to minimize the top-$k$ average loss (which is a relaxation of minimizing the max loss) as follows:

$$l(k, \theta) = \left[ \frac{1}{k} \sum_{i=1}^{2N} [\mathcal{L}_{ctr}(\tilde{\boldsymbol{x}}_i; \theta) - \lambda(k, \theta)]_+ + \lambda(k, \theta) \right],$$

where $\lambda(k, \theta)$ is the $k$-th largest contrastive loss among $\{\mathcal{L}_{ctr}(\tilde{\boldsymbol{x}}_i; \theta)\}_{i=1}^{2N}$. Here a straightforward way is to directly apply the hinge function on contrastive loss during training. The problem with such method is that, since we treat all samples other than the reference as negative, we would inevitably introduce false negative pairs in (1), which makes contrastive loss not representative enough of downstream classification loss. Besides, when the training datasets are imbalanced w.r.t. label (Larson et al., 2016; Dua and Graff, 2017), we are more likely to obtain false negative pairs for part of training samples, exacerbating the sampling bias of contrastive learning. To mitigate such problem, we instead consider the validation stage and use average top-$k$ validation loss as the training objective. Let $f_\theta$ be the contrastive encoder with parameter $\theta$ and $g_\omega$ be the linear layer during validation with parameter $\omega$, our training objective can be formulated as

$$l^{\mathrm{val}}(k, \theta, \omega) = \left[ \frac{1}{k} \sum_{j=1}^{M} \left[ \mathcal{L}_{cls}\left(g_\omega(f_\theta(\boldsymbol{x}_j)), \boldsymbol{y}_j\right) - \lambda^{\mathrm{val}}(k, \theta, \omega) \right]_+ + \lambda^{\mathrm{val}}(k, \theta, \omega) \right], \tag{2}$$

where $\lambda^{\mathrm{val}}(k, \theta, \omega)$ is the $k$-th largest classification loss among validation samples $\{\mathcal{L}_{cls}\left(g_\omega(f_\theta(\boldsymbol{x}_j)), \boldsymbol{y}_j\right)\}_{j=1}^M$. As the update of contrastive encoder affects both training and validation, we want to learn a weight assignment $\{v_i\}_{i=1}^{2N}$ for training samples such that by minimizing the weighted contrastive loss we also minimizes the average top-$k$ validation loss:

$$
\begin{aligned}
\theta^*(v) &= \arg\min_\theta \frac{1}{2N}\left[\sum_{i=1}^{2N} v_i \mathcal{L}_{ctr}(\tilde{\boldsymbol{x}}_i; \theta)\right], \\
v^*, \omega^* &= \arg\min_{v\geq 0, \omega} l^{\mathrm{val}}(k, \theta^*(v), \omega).
\end{aligned}
\tag{3}
$$

## 3.4 Weight Approximation

Directly calculating the optimal weight in (3) can be very expensive. In this subsection, we introduce a surrogate weight based on the gradient similarity between the contrastive loss and the validation loss.

For the simplicity of notation, we omit the parameters in the losses. That is, at iteration $t$, we denote the contrastive loss in (1) as $l_{t,i} = \mathcal{L}_{ctr}(\tilde{\boldsymbol{x}}_i; \theta)$ and denote the validation loss in (2) as $l_t^{\mathrm{val}}$.

As the latent representation from the encoder $f_\theta$ is related with both $l_{t,i}$ and $l_t^{\mathrm{val}}$, we can assign surrogate of weights $v$ by considering the similarity between the gradients on contrastive loss and validation loss. We would expect higher weights on training samples whose gradient directions $\nabla_\theta l_{t,i}$ are closer to that of $\nabla_\theta l_t^{\mathrm{val}}$, such that the weighted update of encoder better approximates the expected update under $l_t^{\mathrm{val}}$. Inspired by Ren et al. (2018), we use a simple approximation of the optimal weight based on the inner product between gradients:

$$
u_{t,i} = \left(\nabla_\theta l_t^{\mathrm{val}}\right)^\top \nabla_\theta l_{t,i}.
\tag{4}
$$

To avoid negative contrastive loss, we apply ReLU function on $u_{t,i}$ and normalize the weights on each augmented mini-batch $\{\tilde{\boldsymbol{x}}_i, 1 \leq i \leq 2n\}$ that is sampled from training set:

$$
\begin{aligned}
\hat{v}_{t,i} &= \max\left(u_{t,i}, 0\right), \\
v_{t,i} &= \frac{2n\hat{v}_{t,i}}{\sum_{i'=1}^{2n} \hat{v}_{t,i'} + \delta\left(\sum_{i'=1}^{2n} \hat{v}_{t,i'}\right)},
\end{aligned}
\tag{5}
$$

where $\delta(r) = 1 \iff r = 0$, and $v_{t,i}$ is the optimal weight for the $i$-th training sample at $t$-iteration

Our method effectively utilizes the gradient information of training validation samples and computes the similarity between the gradient directions as the surrogate weight. Intuitively, if one training sample provides similar direction as samples of top-$k$ validation loss, then this training sample should be upweighted, and otherwise downweighted. In this way, unlike training the encoder only on validation set, our method tries to approximate a distribution of training set such that the modified distribution is beneficial to our optimization objective defined in (3).

## 3.5 Training Algorithm

The gradients in (4) can be easily computed with automatic differentiation. Our training procedure is as follows: we first pre-train the encoder to minimize the classification loss on the small validation set following the method by (Khosla et al., 2020). We then use this pre-trained model as initialization and train the encoder on the training set. During each iteration, we first compute the contrasitve loss for each sample in the forward pass, and fine-tune the linear layer with encoder frozen to compute the gradients and weights in (4) and (5), and the encoder is then updated according to the weighted gradient descent. We summarize the detailed training process in Algorithm 1.

---

**Algorithm 1** Optimization Algorithm

---

Pre-train the encoder $f_\theta$ on the validation set $\{(\boldsymbol{x}_j^{\text{val}}, y_j^{\text{val}}), 1 \le j \le M\}$;
**for** $t = 0, 1, \ldots, T-1$ **do**

    1. Sample a mini-batch of training samples of size $n$, apply random augmentation on each sample twice and get a training set $\{\tilde{\boldsymbol{x}}_i, 1 \le i \le 2n\}$;

    2. Calculate contrastive loss $\{\mathcal{L}_{ctr}(\tilde{\boldsymbol{x}}_i; \theta)\}_{i=1}^{2n}$ as in (1), denote it as $\{l_{t,i}\}_{i=1}^{2n}$;

    3. Freeze $f_\theta$ and fine-tune the linear layer $g_\omega$ on validation set;

    4. Calculate validation loss $l^{\text{val}}(k, \theta, \omega)$ as in (2), denote it as $l_t^{\text{val}}$;

    5. Update $\hat{v}_{t,i} = \max\left((\nabla_\theta l_t^v)^\top \nabla_\theta l_{t,i}, 0\right)$;

    6. Update $v_{t,i} = \frac{2n\hat{v}_{t,i}}{\sum_{i'=1}^{2n} \hat{v}_{t,i'} + \delta\left(\sum_{i'=1}^{2n} \hat{v}_{t,i'}\right)}$, where $\delta(r) = 1 \iff r = 0$;

    7. Update $\theta_{t+1} = \theta_t - \frac{1}{2n} \nabla_\theta \sum_{i=1}^{2n} v_{t,i} l_{t,i}$;

**end for**

---

### 3.6 Convergence Proof

We now discuss the convergence of our algorithm. Although the convergence of SGD has been widely studied (Gower et al., 2019; Nguyen et al., 2018), the convergence of our algorithm is non-trivial given that our training algorithm involves two different loss functions and that the validation loss involves an extra linear layer compared with contrastive loss. We show theoretically that under some mild conditions, our method converges to the critical point, and the detailed proof can be found in the Appendix. Before we state the convergence of our algorithm, we first state the assumptions we need to prove the convergence:

**Assumption 3.1.** We have the following two assumptions.

    **A.** The partial derivative of validation loss $l^{\text{val}}$ with respect to $\theta$ is Lipschitz continuous with constant $L$, i.e., $\nabla_{\omega\theta}^2 l^{\text{val}}$ and $\nabla_{\theta\theta}^2 l^{\text{val}}$ are upper-bounded by $L$.

    **B.** The contrastive loss $l$ has $\sigma$-bounded gradients w.r.t. $\theta$.

And we can prove the convergence of our method under bounded learning rates $\alpha_1$ and $\alpha_2$.

**Theorem 3.2.** *Under Assumption 3.1, at iteration $t$, let the learning rate of contrastive encoder $f$ satisfies $\alpha_{1,t} \le \frac{4\sigma^2 L \sum_i \beta_{t,i}^2}{n \sum_i (\beta_{t,i}^2 - 2\gamma_{t,i}\beta_{t,i})}$, and the learning rate of linear classifier satisfies $\alpha_{2,t} \le \min\left(\frac{2}{L}, \frac{\sum_i \beta_{t,i}^2}{L \sum_i \gamma_{t,i}\beta_{t,i}}\right)$, where*

$$\gamma_{t,i} = \|\nabla_\omega l_t^{val}\| \|\nabla_\theta l_{t,i}\|, \quad \beta_{t,i} = \left((\nabla_\theta l_{t,i})^\top \nabla_\theta l_t^{val}\right),$$

*then the validation loss will monotonically decrease until convergence.*

## 4 Experiments

### 4.1 Experimental Setup

We validate our model on three benchmark datasets. Details of the datasets are as follows:

- **CelebA** (Liu et al., 2015): The dataset contains 202,599 face images, each of resolution $178 \times 218$, with 40 binary attributes. We perform two different tasks on CelebA dataset: gender classification with age as sensitive attribute, and attractiveness classification with gender as sensitive attribute.

- **Adult** (Dua and Graff, 2017) : The dataset contains 65,123 samples with 14 attributes and one binary label indicating if an individual's incoming exceeds $50K$. For Adult dataset, the goal is to predict whether an individual's income exceeds $50K$, and the sensitive attributes are gender and race.

- **COMPAS** (Larson et al., 2016): The dataset contains 7,215 samples with 11 attributes. As with previous works on fairness (Zafar et al., 2017), we only select black and white

individuals in COMPAS dataset, and the modified dataset contains 6,150 samples. For COMPAS dataset, the goal is to predict whether a defendant reoffends within two years, and the sensitive attributes are sex and race.

We implement our method in PyTorch 1.10.1 with one NVIDIA RTX-3090 GPU. We use accuracy to evaluate classification performance, and evaluate fairness using two widespread metrics: disparate impact (Kamiran and Calders, 2012), equalized odds (Hardt et al., 2016). For our method, we implement the contrastive encoder using the structure from (Chen et al., 2020) and the linear classifier using logistic regression. Our method does not require sensitive information in either training or validation set. Labels are available on validation set, but not on training set. We compare our method with six related methods:

- **Fully supervised baseline**: Fully supervised network with same structure as (Chen et al., 2020) and without fairness constraints. Labels are available on both training and validation set. The difference between our method and fully supervised baseline is that the fully supervised baseline are trained with supervised contrastive loss (Khosla et al., 2020) on training and validation set.

- **Contrastive learning baseline**: Contrastive learning method (Chen et al., 2020) without fairness constraints. Labels are only available on validation set. The difference between our method and contrastive learning baseline is that the baseline does not impose fairness constraints during training.

- **Distributionally robust optimization (DRO)**: contrastive learning baseline with distributionally robust optimization (Hashimoto et al., 2018). The difference between our method and DRO is that DRO is applied on contrastive loss during training and on validation loss during validation.

- **Adversarially reweighted learning (ARL)**: contrastive learning baseline with adversarially reweighted learning (Lahoti et al., 2020). The difference between our method and ARL is that ARL is only applied on validation loss during fine-tune. The adversarial network is chosen as one linear layer as suggested by the authors (Lahoti et al., 2020).

- **Postprocessing**: contrastive learning baseline with postprocessing (Hardt et al., 2016). The fairness constraint is equalized odds. The difference between our method and postprocessing is that the latter requires sensitive information in the validation set to adjust the classification threshold w.r.t. the fairness constraint.

- **TAC**: contrastive learning baseline with TAC (Hwang et al., 2020). The sensitive information and label are available on both training and validation set. The difference between our method and TAC is that TAC is applied on fully supervised baseline.

We repeat experiments on each dataset three times and report the average results and in each repetition we randomly spilt data into 64% training data, 16% validation data and 20% test data. All the methods evaluated are trained and tested on the same data partitions each time. All hyperparameters are tuned to find the best validation accuracy. The values of hyperparameter in our method are set by performing cross-validation on training data in the value range of 0.1 to 10. The hyperparameters for the comparing methods are tuned as suggested in the original paper (Hardt et al., 2016; Hashimoto et al., 2018; Hwang et al., 2020).

### 4.2 Experimental Results

Classification results are shown in Tables 1-6. Compared with other state-of-the-art methods on fairness without demographics, our method achieves best or comparable performance on all three datasets regarding both classification accuracy and fairness when we evaluate fairness w.r.t. different sensitive groups. Unlike DRO or ARL which only applies fairness constraints during either training or validation, our method effectively applies fairness constraints on classification during self-supervised training. In this way, our method mitigates the problem that contrastive loss during training might not be representative enough for downstream classification loss. Besides, since we directly apply weight to training samples, we are able to directly modify the update of encoder with supervised information, instead of merely adjusting the update of linear layer during fine-tune.

We also compare our method with conventional fairness methods where the sensitive information is available. Compared with postprocessing, although our method does not require sensitive information

Table 1: Results on the COMPAS dataset with sex as sensitive attribute. Postprocessing and TAC require demographic information. We evaluated Postprocessing and TAC when providing different demographic information (shown in the parenthesis). Lower Disparate Impact and lower Equalized Odds shows more fair results.

| | Methods | Accuracy (%) | Disparate Impact (%) | Equalized Odds (%) |
|---|---|---|---|---|
| Methods with Correct Demographics | Postprocessing (gender) | 63.42±0.54 | 12.31±1.37 | 10.65±3.91 |
| | TAC (gender) | 63.47±0.67 | 6.42±2.56 | 5.03±1.32 |
| Methods with Wrong Demographics | Postprocessing (race) | 63.03±0.74 | 17.64±1.91 | 17.67±2.41 |
| | TAC (race) | 63.82±0.86 | 16.64±1.07 | 16.68±2.23 |
| Methods without Demographics | Fully supervised baseline | **65.23±0.84** | 19.17±4.86 | 20.25±5.19 |
| | Contrastive learning baseline | 64.02±0.93 | 19.23±2.49 | 20.63±5.39 |
| | DRO | 62.21±0.62 | 18.82±3.48 | 16.63±3.21 |
| | ARL | 63.23±0.51 | 17.62±3.63 | 15.37±2.73 |
| | **Our method** | 63.67±0.55 | **17.53±1.90** | **13.34±2.60** |

Table 2: Results on the COMPAS dataset with race as sensitive attribute.

| | Methods | Accuracy (%) | Disparate Impact (%) | Equalized Odds (%) |
|---|---|---|---|---|
| Methods with Correct Demographics | Postprocessing (race) | 63.03±0.74 | 15.52±2.47 | 17.62±3.05 |
| | TAC (race) | 63.82±0.86 | 5.24±2.14 | 4.73±3.35 |
| Methods with Wrong Demographics | Postprocessing (sex) | 63.42±0.54 | 20.41±1.84 | 30.32±3.42 |
| | TAC (sex) | 63.47±0.67 | 19.83±1.59 | 29.72±3.60 |
| Methods without Demographics | Fully supervised baseline | **65.23±0.84** | 22.32±2.94 | 38.37±7.23 |
| | Contrastive learning baseline | 64.02±0.93 | 22.23±2.66 | 38.34±7.38 |
| | DRO | 62.21±0.62 | 21.41±3.89 | 30.43±2.64 |
| | ARL | 63.23±0.51 | 21.37±1.71 | 29.46±3.53 |
| | **Our method** | 63.67±0.55 | **17.46±3.73** | **24.43±3.18** |

Table 3: Results on the Adult dataset with gender as sensitive attribute.

| | Methods | Accuracy (%) | Disparate Impact (%) | Equalized Odds (%) |
|---|---|---|---|---|
| Methods with Correct Demographics | Postprocessing (gender) | 82.21±1.07 | 13.01±2.48 | 10.17±2.64 |
| | TAC (gender) | 84.12±1.53 | 12.17±1.97 | 8.23±3.45 |
| Methods with Wrong Demographics | Postprocessing (race) | 82.07±1.21 | 16.01±2.13 | 20.21±2.55 |
| | TAC (race) | 83.64±1.68 | 18.83±3.67 | 21.44±2.14 |
| Methods without Demographics | Fully supervised baseline | **85.53±0.94** | 19.30±1.43 | 23.34±2.57 |
| | Contrastive learning baseline | 84.11±1.14 | 19.80±2.36 | 22.82±2.17 |
| | DRO | 81.37±2.26 | 16.21±1.87 | 19.43±3.64 |
| | ARL | 82.31±1.12 | 15.82±3.69 | 18.25±2.67 |
| | **Our method** | 82.76±0.73 | **14.81±1.47** | **17.63±1.64** |

on validation set, our method still achieves comparable performance. Compared with TAC, although our method does not require labels on training set or sensitive information on training and validation set, our method still achieves comparable performance regarding accuracy on all three datasets and comparable performance regarding disparate impact on Adult and CelebA datasets. Besides, we also show results when conventional fairness methods are applied with wrong sensitive information. We observe that when there exists a high correlation between different sensitive groups, even wrong sensitive information can play as alternative demographics information to help fairness (compared fairness without demographics methods). However, since the distributional disparity regarding different sensitive attributes are not identical, as shown in the tables, fairness methods with certain predefined sensitive information does not always guarantee fairness under another predefined sensitive groups. On Adult and CelebA dataset, both postprocessing and TAC applied with wrong sensitive

Table 4: Results on the Adult dataset with race as sensitive attribute.

| | Methods | Accuracy (%) | Disparate Impact (%) | Equalized Odds (%) |
|---|---|---|---|---|
| Methods with Correct Demographics | Postprocessing (race) | 82.07±1.21 | 13.34±2.14 | 11.15±3.62 |
| | TAC (race) | 83.64±1.68 | 13.32±2.43 | 9.14±2.25 |
| Methods with Wrong Demographics | Postprocessing (gender) | 82.21±1.73 | 15.17±1.57 | 20.21±2.24 |
| | TAC (gender) | 84.12±1.53 | 14.17±1.59 | 20.65±3.48 |
| Methods without Demographics | Fully supervised baseline | **85.53±0.94** | 13.84±2.54 | 22.33±4.69 |
| | Contrastive learning baseline | 84.11±1.14 | 13.81±1.27 | 22.30±3.69 |
| | DRO | 81.37±2.26 | 13.52±1.67 | 17.35±2.04 |
| | ARL | 82.31±1.12 | 13.34±2.32 | **16.43±2.45** |
| | **Our method** | 82.76±0.73 | **12.21±1.59** | 16.46±2.21 |

Table 5: Results on the CelebA dataset with gender as sensitive attribute and attractive as label.

| | Methods | Accuracy (%) | Disparate Impact (%) | Equalized Odds (%) |
|---|---|---|---|---|
| Methods with Correct Demographics | Postprocessing (gender) | 78.32±0.87 | 11.24±1.88 | 8.67±2.34 |
| | TAC (gender) | 79.32±0.61 | 13.21±1.67 | 10.23±2.96 |
| Methods with Wrong Demographics | Postprocessing (age) | 77.43±1.83 | 14.01±2.56 | 18.42±1.60 |
| | TAC (age) | 78.82±0.71 | 17.31±2.68 | 19.63±2.23 |
| Methods without Demographics | Fully supervised baseline | **80.43±1.62** | 18.62±3.29 | 22.37±5.82 |
| | Contrastive learning baseline | 79.13±0.57 | 18.21±4.03 | 20.64±5.45 |
| | DRO | 76.38±2.66 | 15.33±3.09 | 17.61±4.43 |
| | ARL | 76.43±1.37 | 14.44±2.19 | 16.83±2.76 |
| | **Our method** | 77.63±0.79 | **14.32±1.89** | **16.17±1.97** |

Table 6: Results on the CelebA dataset with age as sensitive attribute and gender as label.

| | Methods | Accuracy (%) | Disparate Impact (%) | Equalized Odds (%) |
|---|---|---|---|---|
| Methods with Correct Demographics | Postprocessing (age) | 86.83±0.86 | 11.17±1.59 | 8.13±3.03 |
| | TAC (age) | 88.12±0.92 | 9.45±2.09 | 5.27±2.48 |
| Methods with Wrong Demographics | Postprocessing (smiling) | 86.32±0.72 | 14.01±1.28 | 12.67±2.15 |
| | TAC (smiling) | 87.76±0.96 | 14.33±2.93 | 12.25±1.75 |
| Methods without Demographics | Fully supervised baseline | **89.74±0.84** | 16.75±4.85 | 14.44±4.80 |
| | Contrastive learning baseline | 87.43±0.84 | 16.25±2.53 | 14.43±4.93 |
| | DRO | 72.43±2.63 | 15.21±1.73 | 13.44±2.34 |
| | ARL | 85.54±0.73 | 14.67±3.59 | 12.59±1.34 |
| | **Our method** | 86.93±0.72 | **11.34±2.50** | **10.82±2.37** |

information achieve trivial improvement in fairness. In comparison, our method without predefined sensitive information works better under different sensitive attributes.

### 4.3 Trade-Off between Fairness and Accuracy

We move on to discuss the fairness-accuracy trade-off with Pareto frontier. Kim et al. (2020) propose to characterize the trade-offs in group fairness via the fairness-confusion tensor $z$ composed of eight elements, and the majority of group fairness constraints can be modeled with specific fairness matrices. By solving the optimization problem of $z$ with model-specific constraints and linear or quadratic fairness regularization terms, we can obtain a model-specific Pareto frontier indicating the best fairness-accuracy trade-off any methods could achieve under such classifier, and smaller distance to the frontier indicates better fairness-accuracy trade-off, and methods lie right on the curve indicates an ideal classifier. In Fig. 1, the blue curve shows the model-specific frontier, where the model is chosen as contrastive learning baseline. Compared with DRO and ARL, our method lies closer to the frontier and achieves better fairness-accuracy trade-off.

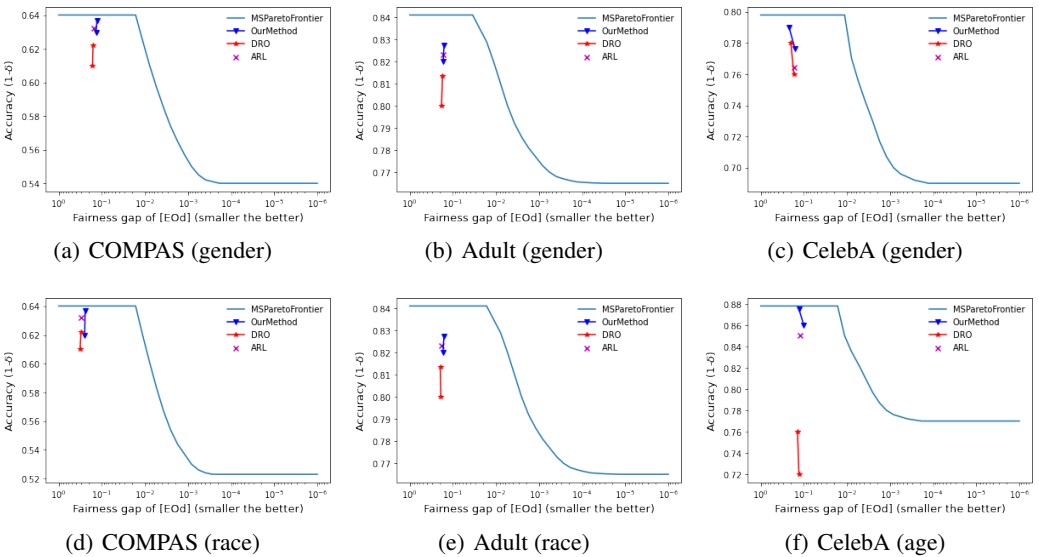

Figure 1: Pareto frontier on Adult, CelebA and COMPAS dataset. The MS Pareto Frontier is plotted by post-processing the contrastive learning baseline.

## 5 Conclusion

Improving fairness without accessing sensitive attributes has been a challenging task. In this paper, we propose a fair contrastive learning method with gradient-based reweighing to learn a fair representation for downstream classification tasks. We consider a semi-supervised setting where a small validation set is available, and use top-$k$ average loss as surrogate fairness constraints. We assign weight based on gradient similarity to apply information of validation loss during self-supervised training. We theoretically prove the convergence of our method, and we show from experiments that our method improves fairness with relatively small sacrifice in classification accuracy across multiple datasets. Future directions of interest includes robustness of our method to noisy validation set, and alternative ways of sampling validation set to further improve fairness.

## Acknowledgements

This work was partially supported by the EMBRIO Institute, contract #2120200, a National Science Foundation (NSF) Biology Integration Institute, Purdue's Elmore ECE Emerging Frontiers Center, and NSF IIS #1955890, IIS #2146091.

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
