# A  Experiments Supplement

## A.1  Hyperparameters Sensitivity

Here we test the sensitivity of model w.r.t. the hyperparameter $\lambda^{\text{val}}$ in (2). Since most loss values falls within the range of $[0.1, 10]$, we evaluate how the model accuracy and fairness change w.r.t. different cutoff values $\lambda^{\text{val}}$. Fig. 2 shows the change of accuracy under different cutoff value. Generally, increasing $\lambda^{\text{val}}$ increases the test accuracy, as the distribution of top-$k$ validation samples becomes more similar to that of the whole validation set, and the classifier focuses more on overall validation samples, instead of only hard sample. However, for gender classification under CelebA dataset, the trade-off between $\lambda^{\text{val}}$ and accuracy is not very clear; and we suspect that under such scenario, focusing on hard samples does not harm the performance of easy samples, and thus benefits the classifier.

Fig. 5 shows the change of fairness (equalized odds) under different cutoff value. The overall equalized odds shows an increasing trend as $\lambda^{\text{val}}$ increases, which validates the effectiveness of Max-Min objective for improving group fairness.

## A.2  Sensitivity of Validation Size

We show the effect of validation size on accuracy and equalized odds in Fig. . As shown in the figures, when the validation size is larger than $10\%$ of training size, the model's performance becomes stable in terms of accuracy and fairness.

# B  Proof of Theorem 3.2

Suppose we have a large unlabeled training set of size $N$ and a small labeled validation set $\{(\boldsymbol{x}_j^{\text{val}}, \boldsymbol{y}_j^{\text{val}}), 1 \leq j \leq M\}$ with $M \ll N$. In each training step, we sample a small mini-batch of size $n (n < N)$ from training set and perform random augmentation twice to obtain a subset $\{\tilde{\boldsymbol{x}}_i, 1 \leq i \leq 2n\}$ and we update the contrastive encoder $f$ with parameter $\theta$. During validation, we freeze the contrastive encoder and train a downstream linear classifier $g$ with parameter $\omega$ for classification task.

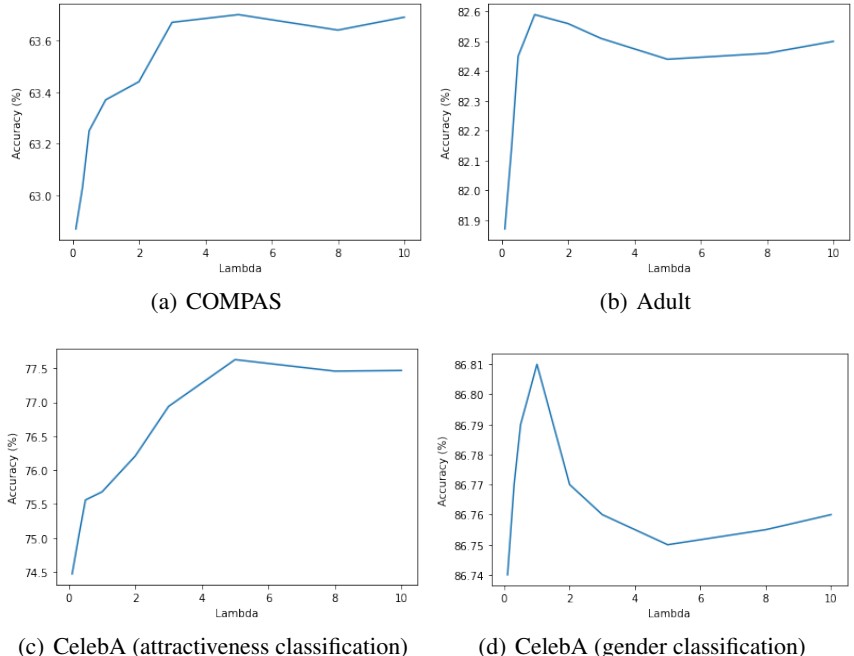

(a) COMPAS

(b) Adult

(c) CelebA (attractiveness classification)

(d) CelebA (gender classification)

Figure 2: Change of accuracy as $\lambda^{\text{val}}$ varies.

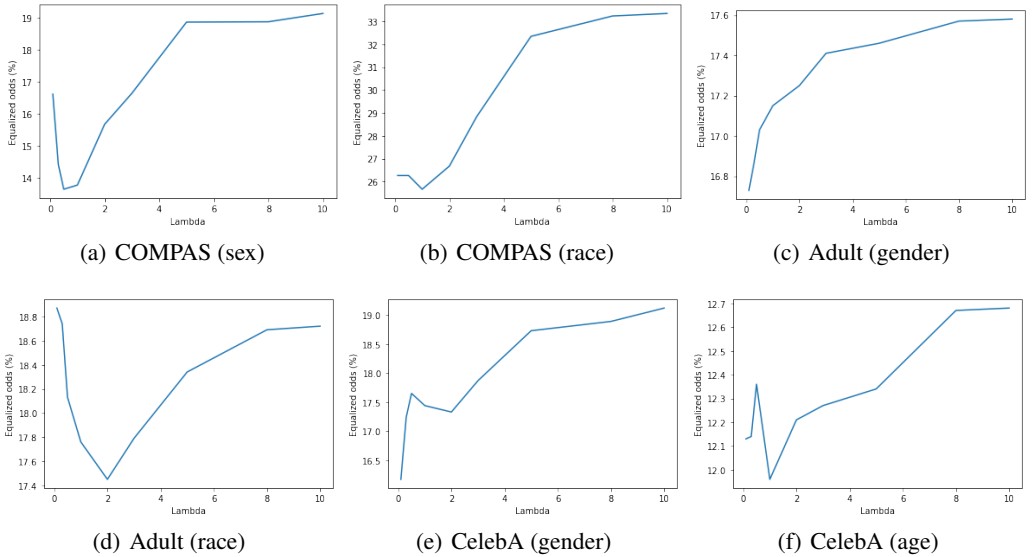

Figure 3: Change of equalized odds as $\lambda^{\mathrm{val}}$ varies.

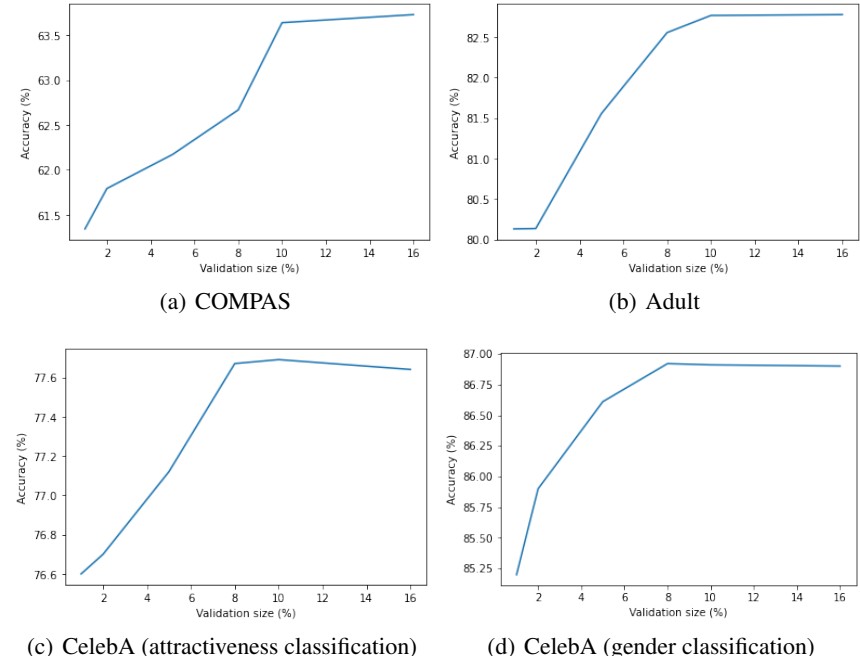

Figure 4: Change of accuracy as validation size varies.

Denote the training (contrastive) loss of the $i$-th data at iteration $t$ as

$$l_{t,i} = -\log \frac{\exp\left(\mathrm{sim}\left(\boldsymbol{z}_{t,i}, \boldsymbol{z}_{t,j}\right)/\tau\right)}{\sum_{h \neq i} \exp\left(\mathrm{sim}\left(\boldsymbol{z}_{t,i}, \boldsymbol{z}_{t,h}\right)/\tau\right)}, \tag{6}$$

where $\boldsymbol{z}_{t,i} = f(\tilde{\boldsymbol{x}}_i; \theta_t)$ is the output of contrastive encoder $f$ for the $i$-th training data at iteration $t$, $\mathrm{sim}\left(\boldsymbol{z}_{t,i}, \boldsymbol{z}_{t,j}\right) = \frac{\boldsymbol{z}_{t,i}\boldsymbol{z}_{t,j}}{\|\boldsymbol{z}_{t,i}\|\|\boldsymbol{z}_{t,j}\|}$ is the cosine similarity and $\tau$ is the temperature scaling hyper-parameter.

For ease of exposition, here we consider binary classification (i.e., $y_j^{\mathrm{val}}|_{j=1}^M \in \{0,1\}$). Our proof can easily extend to the multi-class classification scenario, since the only difference is that we will use

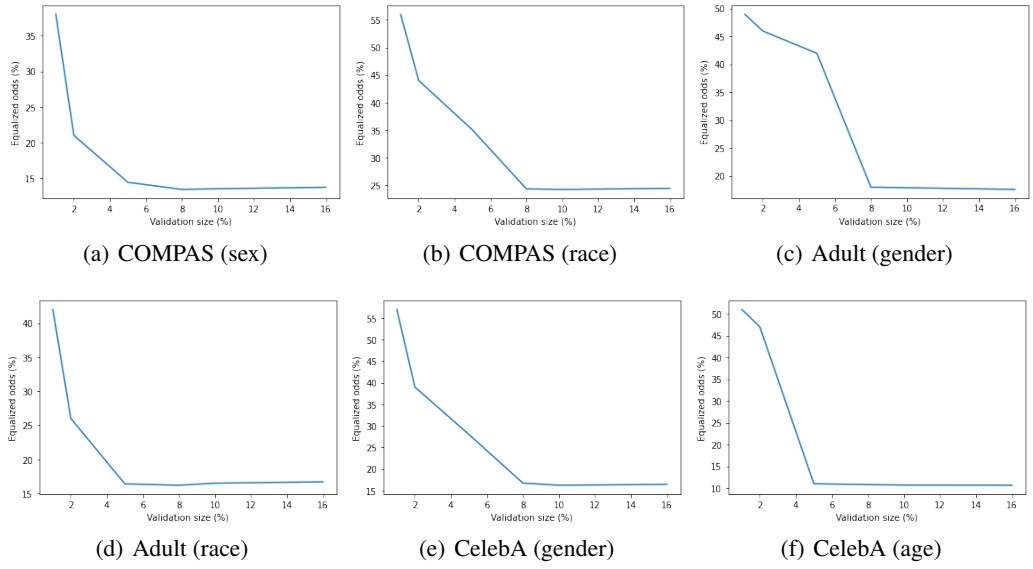



(a) COMPAS (sex)      (b) COMPAS (race)      (c) Adult (gender)

(d) Adult (race)      (e) CelebA (gender)      (f) CelebA (age)



Figure 5: Change of equalized odds as validation size varies.

cross entropy loss (instead of binary cross entropy loss in (7) below). Denote $\lambda_t$ as the $k$-th largest validation loss at iteration $t$. The top-$k$ average validation (classification) loss is represented as

$$l_t^{\text{val}} = \left[ \frac{1}{k} \sum_{j=1}^{M} \left[ \left( -y_j^{\text{val}} log(\hat{y}_{t,j}^{\text{val}}) - (1 - y_j^{\text{val}}) log(1 - \hat{y}_{t,j}^{\text{val}}) \right) - \lambda_t \right]_+ + \lambda_t \right], \tag{7}$$

where $\hat{y}_{t,j}^{\text{val}} = g(z_j; \omega_t) = \frac{1}{1 + e^{-\omega_t^\top z_{t,j}^{\text{val}}}}$ is the predicted output of the linear classifier, and $z_{t,j}^{\text{val}} = f(x_j^{\text{val}}; \theta_t)$ is the output of the contrastive encoder $f$ for the $j$-th validation data at iteration $t$. For the ease of notation, in the following we drop the parameter $k$ and use $l_t^{\text{val}}$ to represent the top-$k$ average validation loss. In Section A.1 above, we showed the sensitivity of model performance and fairness w.r.t. different cutoff values $\lambda_t$. Increasing $k$ means decreasing cutoff value $\lambda_t$.

Now we are ready to prove the prove the converence in Theorem 3.2.

*Proof.* In each training step, the parameter $\theta$ of contrastive encoder $f$ is updated with mini-batch stochastic gradient descent (SGD) as follows:

$$\theta_{t+1} = \theta_t - \alpha_1 \nabla_\theta \left( \frac{1}{2n} \sum_{i=1}^{2n} v_{t,i} l_{t,i} \right), \tag{8}$$

where $v_{t,i}$ is the weight assigned to training sample $x_i$ at iteration $t$.

During validation, the contrastive encoder is frozen, and the parameter $\omega$ of linear classifier $g$ is updated according to

$$W_{t+1} = W_t - \alpha_2 \nabla_\omega l_t^{\text{val}}. \tag{9}$$

Following the protocol of Ren et al. (2018), we approximate $v_{t,i}$ as

$$u_{t,i} = - \left( \frac{\partial l_t^{\text{val}}}{\partial \omega} \right)^\top \frac{\partial \omega}{\partial \theta} \frac{\partial \theta}{\partial v_{t,i}} \bigg|_{v_{t,i}=0}, \tag{10}$$

$$v_{t,i} = \max\left( u_{t,i}, 0 \right).$$

Here we use the negative derivative of $l_t^{\text{val}}$ with respect to $v_{t,i}$ since our goal is to minimize the validation loss through reweighing, and if $u_{t,i} > 0$, applying this positive weight will increase the

validation loss. By taking the derivative of both sides in (8) w.r.t. $v_{t,i}$, we have

$$\frac{\partial \theta}{\partial v_{t,i}} \propto -\frac{\partial l_{t,i}}{\partial \theta} . \tag{11}$$

Here we can omit the influence of $\alpha_1$ since we normalize the weights on each mini-batch to match the original training step size.

Combining (10) and (11), we can approximate $v_{t,i}$ as

$$v_{t,i} = \max\left(\left(\nabla_\theta l_t^{\text{val}}\right)^\top \nabla_\theta l_{t,i}, 0\right) . \tag{12}$$

Combining (12) and (8), we can derive the following update rule for contrastive encoder parameter:

$$\theta_{t+1} = \theta_t - \frac{\alpha_1}{2n}\left(\sum_{i=1}^{2n} \max\left(\left(\frac{\partial l_t^{\text{val}}}{\partial \theta}\right)^\top \frac{\partial l_{t,i}}{\partial \theta}, 0\right)\nabla_\theta l_{t,i}\right) . \tag{13}$$

Based on Taylor's expansion, we can express the validation loss at $(t+1)$-th iteration as

$$\begin{aligned}
l_{t+1}^{\text{val}} &= l_t^{\text{val}} + \left(\nabla_\theta l_t^{\text{val}}\right)^\top \delta\theta + \left(\nabla_\omega l_t^{\text{val}}\right)^\top \delta\omega + \frac{1}{2}\left(\delta\theta\right)^\top \nabla_{\theta\theta}^2 l_t^{\text{val}}\delta\theta \\
&\quad + \left(\delta\theta\right)^\top \nabla_{\theta\omega}^2 l_t^{\text{val}}(\delta\omega) + \frac{1}{2}(\delta\omega)^\top \nabla_{\omega\omega}^2 l_t^{\text{val}}(\delta\omega) .
\end{aligned} \tag{14}$$

Denote

$$\boldsymbol{r}_t = \left(\sum_{i=1}^{2n}\max\left(\left(\nabla_\theta l_t^{\text{val}}\right)^\top \nabla_\theta l_{t,i}, 0\right)\nabla_\theta l_{t,i}\right). \tag{15}$$

Plugging in the update rules (9) and (13) to (14), we can substitute $\delta\theta$ and $\delta\omega$ with their respective gradient descent form in each iteration, and we have

$$\begin{aligned}
l_{t+1}^{\text{val}} &= l_t^{\text{val}} - \frac{\alpha_1}{2n}\left(\nabla_\theta l_t^{\text{val}}\right)^\top \boldsymbol{r}_t - \alpha_2\left(\nabla_\omega l_t^{\text{val}}\right)^\top\left(\nabla_\omega l_t^{\text{val}}\right) + \frac{\alpha_1^2}{8n^2}\boldsymbol{r}_t^\top \nabla_{\theta\theta}^2 l_t^{\text{val}}\boldsymbol{r}_t \\
&\quad + \frac{\alpha_1\alpha_2}{2n}\boldsymbol{r}_t^\top \nabla_{\theta\omega}^2 l_t^{\text{val}}\nabla_\omega l_t^{\text{val}} + \frac{\alpha_2^2}{2}\left(\nabla_\omega l_t^{\text{val}}\right)^\top \nabla_{WW}^2 l_t^{\text{val}}\nabla_\omega l_t^{\text{val}} .
\end{aligned} \tag{16}$$

Since the validation loss is Lipschitz-smooth, we can replace the second-order derivative $\nabla_{\omega\omega}^2 l_t^{\text{val}}$, $\nabla_{\theta\omega}^2 l_t^{\text{val}}$ and $\nabla_{\theta\theta}^2 l_t^{\text{val}}$ in (16) with constant $L$, and we derive the following inequality

$$\begin{aligned}
l_{t+1}^{\text{val}} &\leq l_t^{\text{val}} + \|\nabla_\omega l_t^{\text{val}}\|^2\left(\frac{\alpha_2^2 L}{2} - \alpha_2\right) + \boldsymbol{r}_t^\top\left(\frac{\alpha_1^2 L}{8n^2}\boldsymbol{r}_t - \frac{\alpha_1}{2n}\nabla_\theta l_t^{\text{val}}\right) + \frac{\alpha_1\alpha_2 L}{2n}\boldsymbol{r}_t^\top \nabla_\omega l_t^{\text{val}} \\
&\leq l_t^{\text{val}} + \|\nabla_\omega l_t^{\text{val}}\|^2\left(\frac{\alpha_2^2 L}{2} - \alpha_2\right) + \boldsymbol{r}_t^\top\left(\frac{\alpha_1^2 L}{8n^2}\boldsymbol{r}_t - \frac{\alpha_1}{2n}\nabla_\theta l_t^{\text{val}}\right) + \frac{\alpha_1\alpha_2 L}{2n}\|\boldsymbol{r}_t\|\|\nabla_\omega l_t^{\text{val}}\| .
\end{aligned} \tag{17}$$

Plugging the definition of $\boldsymbol{r}_t$ in (15) back into (17) we can rewrite (17) as

$$\begin{aligned}
l_{t+1}^{\text{val}} &\leq l_t^{\text{val}} + \|\nabla_\omega l_t^{\text{val}}\|^2\left(\frac{\alpha_2^2 L}{2} - \alpha_2\right) + \frac{\alpha_1^2 L}{8n^2}\sum_{i=1}^{2n}\left(\max\left(\nabla_\theta l_t^{\text{val}}\right)^\top \nabla_\theta l_{t,i}, 0\right)^2 \|\nabla_\theta l_{t,i}\|^2 \\
&\quad - \frac{\alpha_1}{2n}\left(\sum_{i=1}^{2n}\max\left(\left(\nabla_\theta l_t^{\text{val}}\right)^\top \nabla_\theta l_{t,i}, 0\right)\nabla_\theta l_{t,i}^\top \nabla_\theta l_t^{\text{val}}\right) \\
&\quad + \frac{\alpha_1\alpha_2 L}{2n}\left(\sum_{i=1}^{n}\max\left(\left(\nabla_\theta l_t^{\text{val}}\right)^\top \nabla_\theta l_{t,i}, 0\right)\|\nabla_\theta l_{t,i}\|\|\nabla_\omega l_t^{\text{val}}\|\right) ,
\end{aligned} \tag{18}$$

where the third term on the right hand side of the inequality is derived from Jensen's inequality.

For any $r \in \mathbb{R}$, we can derive that $r \cdot \max(r, 0) = \max(r, 0)^2$, thus we can rewrite the fourth and fifth terms on the right hand side of (18) and get:

$$l_{t+1}^{\text{val}} \leq l_t^{\text{val}} + \|\nabla_\omega l_t^{\text{val}}\|^2 \left( \frac{\alpha_2^2 L}{2} - \alpha_2 \right) + \frac{\alpha_1^2 L}{8n^2} \sum_{i=1}^{2n} \left( \max \left( \nabla_\theta l_t^{\text{val}} \right)^\top \nabla_\theta l_{t,i}, 0 \right)^2 \|\nabla_\theta l_{t,i}\|^2$$

$$- \frac{\alpha_1}{2n} \sum_{i=1}^{2n} \max \left( \left( \nabla_\theta l_t^{\text{val}} \right)^\top \nabla_\theta l_{t,i}, 0 \right)^2 \tag{19}$$

$$+ \frac{\alpha_1 \alpha_2 L}{2n} \left( \sum_{i=1}^{2n} \max \left( \left( \nabla_\theta l_t^{\text{val}} \right)^\top \nabla_\theta l_{t,i}, 0 \right)^2 \frac{\|\nabla_\theta l_{t,i}\|\|\nabla_\omega l_t^{\text{val}}\|}{\left( \nabla_\theta l_t^{\text{val}} \right)^\top \nabla_\theta l_{t,i}} \right),$$

Given that the contrastive loss of training data have $\sigma$-bounded gradients w.r.t. $\theta$, and consider the definition of $v_{t,i}$ in (12), we can further simplify (19) as

$$l_{t+1}^{\text{val}} \leq l_t^{\text{val}} + \|\nabla_\omega l_t^{\text{val}}\|^2 \left( \frac{\alpha_2^2 L}{2} - \alpha_2 \right) + \sum_{i=1}^{2n} v_{t,i}^2 \left( \frac{\alpha_1^2 \sigma^2 L}{8n^2} - \frac{\alpha_1}{2n} + \frac{\alpha_1 \alpha_2 L}{2n} \frac{\|\nabla_\omega l_t^{\text{val}}\|\|\nabla_\theta l_{t,i}\|}{\left( \nabla_\theta l_{t,i} \right)^\top \nabla_\theta l_t^{\text{val}}} \right). \tag{20}$$

Consider the last term in (20), which is a summation over $2n$ training samples. Denote $cos(\beta_{t,i}) = \frac{(\nabla_\theta l_{t,i})^\top \nabla_\theta l_t^{\text{val}}}{\|\nabla_\theta l_{t,i}\|\|\nabla_\theta l_t^{\text{val}}\|}$ and split the $n$ training sample indices into two sets $\mathcal{S}_t = \{i | cos(\beta_{t,i}) > 0\}$ and $\bar{\mathcal{S}}_t = \{i | cos(\beta_{t,i}) \leq 0\}$. We can observe that for any sample $x_i$ where $i \in \bar{\mathcal{S}}_t$, we have the sample weight $v_{t,i} = \max \left( \left( \nabla_\theta l_t^{\text{val}} \right)^\top \nabla_\theta l_{t,i}, 0 \right) = 0$. Thus we can omit sample indices in $\bar{\mathcal{S}}_t$ in the last term of (20) and obtain the following

$$l_{t+1}^{\text{val}} \leq l_t^{\text{val}} + \|\nabla_\omega l_t^{\text{val}}\|^2 \left( \frac{\alpha_2^2 L}{2} - \alpha_2 \right)$$

$$+ \sum_{i \in \mathcal{S}_t} \left( \left( \nabla_\theta l_t^{\text{val}} \right)^\top \nabla_\theta l_{t,i} \right)^2 \left( \frac{\alpha_1^2 \sigma^2 L}{8n^2} - \frac{\alpha_1}{2n} + \frac{\alpha_1 \alpha_2 L}{2n} \frac{\|\nabla_\omega l_t^{\text{val}}\|\|\nabla_\theta l_{t,i}\|}{\left( \nabla_\theta l_{t,i} \right)^\top \nabla_\theta l_t^{\text{val}}} \right)$$

$$= l_t^{\text{val}} + \|\nabla_\omega l_t^{\text{val}}\|^2 \left( \frac{\alpha_2^2 L}{2} - \alpha_2 \right) + \sum_{i \in \mathcal{S}_t} \left( \frac{\sigma^2 L}{8n^2} \left( \left( \nabla_\theta l_{t,i} \right)^\top \nabla_\theta l_t^{\text{val}} \right)^2 \right) \alpha_1^2$$

$$- \sum_{i \in \mathcal{S}_t} \left( \frac{1}{2n} \left( \left( \nabla_\theta l_{t,i} \right)^\top \nabla_\theta l_t^{\text{val}} \right)^2 - \frac{\alpha_2 L}{2n} \|\nabla_\omega l_t^{\text{val}}\|\|\nabla_\theta l_{t,i}\| \left( \nabla_\theta l_{t,i} \right)^\top \nabla_\theta l_t^{\text{val}} \right) \alpha_1.$$

Since the learning rates satisfy that $\alpha_{1,t} \leqslant \frac{4\sigma^2 L \sum_i \beta_{t,i}^2}{n \sum_i \left( \beta_{t,i}^2 - 2\gamma_{t,i}\beta_{t,i} \right)}$, and $\alpha_{2,t} \leq \min \left( \frac{2}{L}, \frac{\sum_i \beta_{t,i}^2}{L \sum_i \gamma_{t,i}\beta_{t,i}} \right)$ by definition, where

$$\gamma_{t,i} = \|\nabla_\omega l_t^{\text{val}}\|\|\nabla_\theta l_{t,i}\|, \quad \beta_{t,i} = \left( \left( \nabla_\theta l_{t,i} \right)^\top \nabla_\theta l_t^{\text{val}} \right),$$

it follows that $l_{t+1}^{\text{val}} \leq l_t^{\text{val}}$ for any t. $\qquad \square$