# OpenReview forum: "Self-Supervised Fair Representation Learning without Demographics"
_NeurIPS.cc/2022/Conference — NeurIPS 2022 Accept_

### Official Review · Reviewer_kdXn · 2022-07-09

**Rating:** 6
**Confidence:** 4
**Soundness:** 3 good
**Presentation:** 3 good
**Contribution:** 3 good

**Summary:**

This paper studies the fairness issue in machine learning. It aims to improve fair classification without sensitive information and without labels. Specifically, this paper proposes a novel reweighing-based contrastive learning method to learn a generally fair representation without observing sensitive attributes. The idea of this paper is direct and reasonable, and the experimental results demonstrate the effectiveness of the proposed method.

**Questions:**

	In the line 132, how to obtain α^' under the situation where sensitive attributes are unknown?

	From the line 160 to 163, why is treating all samples as negative a problem? This sentence is not explained clearly. Could the authors give a more detail explanation or an example?


	The authors describe the λ is a hyperparameter in Appendix, but in the equation 2, it looks like the λ is determined by k and k is a hyperparameter. Is the experiment in A.2 conducted on different k?


**Limitations:**

N.A

**Strengths And Weaknesses:**

Strengths:

1.	The paper is clearly written and is easy to follow.

2.	The setting in this paper, without sensitive information and without labels, has high application value and is rarely studied before.

3.	The method in this paper has theoretical guarantee.

4.	Extensive experiments are shown to demonstrate the effectiveness of the proposed framework.

Weakness:

1.	There is a typo in line 160, “… to is to …” —> “… is to …”.

2.	It’s not clear why using top-k average loss.

3.	More ablation study needs to add. It seems like there are requirements for the size of the validation set when using it as a guide. The authors had better to analyze the effect of different size of the validation set.

---

> ### Author Response · Authors · 2022-08-02
> **Response to reviewer kdXn**
>
> We thank the reviewer for the comment. We'll fix the typo in final paper.
>
> * **Why top-k average loss:** Similar to DRO [1] and ARL [2], our main assumption is that disadvantaged group(s) tend to have larger portion of hard samples (i.e. samples that are prone to be wrongly classified), in other words, disadvantaged samples are more likely to have higher classification loss. Therefore, by focusing on hard samples in training set we are able to improve the performance of disadvantaged groups and reduce the discrepancy between statistical features of different sensitive groups. As hard samples are prone to be wrongly classified during training, we choose top-$k$ average validation loss as the objective to improve the performance on hard samples.
>
> * **Ablation study:** We show the effect of validation size on accuracy and equalized odds in in [Fig. 1 and Fig. 2](https://docs.google.com/document/d/1Cumrg2DcYGTPR_4gpEMKrUzJW35EGvUlOni7jRubNx4/edit?usp=sharing). As shown in the figures, when the validation size is larger than $10\%$ of training size, the model's performance becomes stable in terms of accuracy and fairness.
>
> * **Q1:** In Line 132, it is hard to find an accurate estimation of sensitive information $\alpha'$ when only accessing label information. Possible ways to define $\alpha'$ include implicitly sampling $\alpha'$ as the worst-performance subset (DRO [1], ARL [2], our method) and using clustering information as surrogate sensitive information [3].
>
> * **Q2:** The idea of contrastive learning is to encourage the similarity between representations of positive pairs and discourage the similarity between representations of negative pairs, where the negative pairs are randomly sampled on whole training set. However, many datasets are imbalanced w.r.t. label, making this random sampling a problem. For example, Adult dataset contains around $75\%$ samples with negative labels and around $25\%$ samples with positive labels. If we randomly sample points from the training set and use these samples to construct negative pairs, for samples with ground-truth negative label, we are likely to obtain false negative pairs (samples with the same ground-truth label as negative label), and the contrastive loss becomes biased. We'll clarify it in final paper.
>
> * **Q3:** We are sorry for the confusion. Experiments in A.2 are conducted under different $\lambda$. Since different $k$ corresponds to different cutoff value $\lambda$, varying $\lambda$ is equivalent to changing the value of $k$.
>
> **[Reference]**
>
>
> [1] Hashimoto, T., Srivastava, M., Namkoong, H., \& Liang, P. Fairness without demographics in repeated loss minimization. In ICML 2018.
>
> [2] Lahoti, P., Beutel, A., Chen, J., Lee, K., Prost, F., Thain, N., ... \& Chi, E. Fairness without demographics through adversarially reweighted learning. In NeurIPS 2020.
>
> [3] Yan, S., Kao, H. T., \& Ferrara, E. Fair class balancing: Enhancing model fairness without observing sensitive attributes. In CIKM 2020.

---

### Official Review · Reviewer_Xdpb · 2022-07-09

**Rating:** 7
**Confidence:** 4
**Soundness:** 3 good
**Presentation:** 3 good
**Contribution:** 3 good

**Summary:**

This paper introduces a novel problem “Can we improve fair classification without sensitive information and without labels?” To tackle this problem, they propose a reweighing-based contrastive learning method. Then, they provide theoretical proof of the convergence of their model. Finally, the conduct experiments on three datasets to show their proposed method achieves better performance.

**Questions:**

According to the authors, there are weaknesses in this work that have not been addressed. Please refer to the main reviews.

**Limitations:**

1)	Maybe due to the space limitation, the detail of the experiment is not well introduced.
2)	The methods with correct demographics baselines are not the start-of-the-art fairness improved methods.


**Strengths And Weaknesses:**

Strengths:

1)	The motivation for this paper is intuitive and well explained. Meanwhile, the paper is well organized.
2)	The authors consider a semi-supervised setting where a small validation set is available, which has strong research significance.
3)	The authors propose a novel contrastive learning method and can solve their proposed question well.
4)	The authors provide theoretical proof of the convergence of their method.
5)	The authors validate the effectiveness of our method in improving fairness on three benchmark datasets.

However, I still have some concerns:

1)	The authors consider a semi-supervised setting compared with DRO and ARL. However, there are many semi-supervised methods in classification. Are they can combined with DRO or ARL? why authors consider using contrastive learning methods to improve the semi-supervised setting?
2)	In experiment, the authors compare their methods with DRO and ARL which are under a fully supervised setting. However, their methods get better performance in both accuracy and fairness metrics. I'm a little skeptical about this result. And in experiment analysis, the authors don’t analysis the reasons. What’s the experiment setting of DRO and ARL? Besides, the authors compared their methods with methods with correct demographics. I understand this is not the key point of this paper. But why the authors just compare with Postprocessing and TAC instead of the start-of-the-art fairness improved methods?

---

> ### Author Response · Authors · 2022-08-02
> **Response to reviewer Xdpb**
>
> We thank the reviewer for the comment.
>
> * **Choice of contrastive learning:** We agree that DRO [1] and ARL [2] might indeed be combined with other semi-supervised learning methods. However, in recent works, contrastive learning has been shown to achieve comparable performance than fully supervised baselines for classification tasks with only partially available label information, and it has also been shown to outperform other semi-supervised learning baselines [3,4]. Thus, we choose the promising contrastive learning method to conduct semi-supervised learning.
>
> * **Experimental setup:** We are sorry that the detail of the experiment is not well introduced due to page limit. Both DRO [1] and ARL [2] are trained under the same label information as our method, i.e., only a small labelled validation set is available during training. DRO [1] is applied on contrastive loss and during fine-tuning linear layer, and ARL is only applied during fine-tuning linear layer, as ARL [2] cannot be applied to the contrastive loss since it requires label information.
>
> * **Fairness baselines:** Thank you for the question. We clarify that our main focus of experiments is to compare our method with DRO [1] and ARL [2], the two most related works on fairness without demographics. For fairness with sensitive information, Postprocessing [5] is a very classic method in fairness, and TAC [6] is an in-processing method, and our method is in-processing. We compare our method with the two works to show that on some datasets, our method can even perform comparably than fairness methods with correct demographics, and that on the three datasets our method perform better or comparably than fairness methods with wrong demographics. We will include more state-of-the-art baselines in final paper.
>
> **[Reference]**
>
> [1] Hashimoto, T., Srivastava, M., Namkoong, H., \& Liang, P. Fairness without demographics in repeated loss minimization. In ICML 2018.
>
> [2] Lahoti, P., Beutel, A., Chen, J., Lee, K., Prost, F., Thain, N., ... \& Chi, E. Fairness without demographics through adversarially reweighted learning. In NeurIPS 2020.
>
> [3] Chen, T., Kornblith, S., Norouzi, M., \& Hinton, G. A simple framework for contrastive learning of visual representations. In ICML 2020.
>
> [4] Verma, V., Luong, T., Kawaguchi, K., Pham, H., \& Le, Q. Towards domain-agnostic contrastive learning. In ICML 2021.
>
> [5] Hardt, M., Price, E., \& Srebro, N. Equality of opportunity in supervised learning. In NeurIPS 2016.
>
> [6] Hwang, S., Park, S., Lee, P., Jeon, S., Kim, D., \& Byun, H. Exploiting transferable knowledge for fairness-aware image classification. In ACCV 2020.

---

> > ### Comment · Reviewer_Xdpb · 2022-08-08
> > **Thank you for the response**
> >
> > Thank you for your response! The answers clarify most of my questions about the paper. I will keep my score. Thank you.

---

> > > ### Author Response · Authors · 2022-08-08
> > > **Thank you**
> > >
> > > Thanks very much for your recognition of our work and your valuable suggestions! We really appreciate all the time and effort you put into reviewing our paper.

---

### Official Review · Reviewer_4iqE · 2022-07-11

**Rating:** 5
**Confidence:** 3
**Soundness:** 3 good
**Presentation:** 3 good
**Contribution:** 3 good

**Summary:**

This work introduces a contrastive method that reweighs training samples based on objective loss gradients. The proposed method is very interesting and takes the form of a training procedure (instead of defining more models or editing existing ones), which means that it is particularly suitable for reuse by non-experts. The paper is well-organized, but I think it needs a revision to improve explanations and make in-depth arguments that would help explain why the proposed approach works. The experiment methodology is sound and results are convincing compared to existing alternatives.

**Questions:**

Please see my comments in Strengths and Weaknesses section.

**Limitations:**

Please see my comments in Strengths and Weaknesses section.

**Strengths And Weaknesses:**

- The greatest shortcoming of the paper is that, in section 3.4, the explanation of why the particular weighting strategy was chosen is too brief and thus feels rather arbitrary. In my opinion, a couple of paragraphs are needed to corroborate why this strategy is expected to work. Two particular concerns I would personally like to see addressed (in addition to other explanations of why following gradients can improve fairness) is that gradients may be similar, but integrating over them can create huge deviations in training outcomes in terms of loss and corresponding accuracy (the authors do a nice job in showing that training converges, but is this enough?) and why should a distribution be approximated by relu-ing data instead of normalizing their sigmoid transformation, which is more typical in machine learning.
- The paper measures specific types of fairness, but it is not explained why the proposed approach can address these in particular. Note that addressing multiple types of fairness simultaneously perfectly is known to be an impossible task unless very specific conditions are satisfied [Kleinberg et al. (2016)], which begs the question what kind of fairness objective the proposed approach is particularly good at addressing.

Minor comments:
- In section 2, please give a brief explanation of the main idea of contrastive learning.
- "Here a straightforward way to is to" > incomplete sentence
- "The problem with such method is," > "The problem with such a method is that,"
- It could be helpful to refer to [1] for better positioning the proposed approach in the context of AI bias literature.

[1] Ntoutsi, E., Fafalios, P., Gadiraju, U., Iosifidis, V., Nejdl, W., Vidal, M. E., ... & Staab, S. (2020). Bias in data‐driven artificial intelligence systems—An introductory survey. Wiley Interdisciplinary Reviews: Data Mining and Knowledge Discovery, 10(3), e1356.

---

> ### Author Response · Authors · 2022-08-02
> **Response to reviewer 4iqE**
>
> We thank the reviewer for the comment. We'll fix the typos and include the suggested reference in final version.
>
> * **Weighting strategy:** Using contrastive loss alone could create certain deviations in terms of loss and accuracy, as during training we would inevitably sample false negative pairs. Instead, we consider a small labelled validation set as reference to perform reweighing, where we use the inner product between the gradient of top-$k$ average validation loss and the gradient of contrastive loss for weight assignment to minimize the top-$k$ average validation loss. In this way, we try to make sure that the minimization of contrastive loss also helps the decrease of top-$k$ average validation loss in order to mitigate the deviation of loss and accuracy. Besides, we show Theorem 3.2 that as long as the learning rate satisfies the constraint, we are able to make sure that during each training iteration the top-$k$ average validation loss monotonically decreases as the weighted contrastive loss is minimized.
>
> * **Distribution approximation by relu-ing data:** We use inner product between the gradient of top-$k$ average validation loss and the gradient of contrastive loss as surrogate weight. In this way, we apply ReLU function on unlabelled data for two reasons: 1) we want to mitigate negative training loss (i.e., negative sample weight) to avoid unstable training process; 2) by applying ReLU function on training set we are able to make sure that the reweighted training samples have positive contributions to the minimization of top-$k$ average validation loss, thus improves fairness.
>
> * **Fairness objective:** Similar to DRO [1] and ARL [2], our main assumption is that disadvantaged group(s) tend to have larger portion of hard samples (i.e. samples that are prone to be wrongly classified), in other words, disadvantaged samples are more likely to have higher classification loss. Therefore, by focusing on hard samples in training set we are able to improve the performance of disadvantaged groups. In this way, by minimizing top-$k$ average validation loss, one directly related fairness objective is equalized odds (assume binary label $y$ and binary sensitive attribute $A$):
>
> \begin{equation}
> \begin{aligned}
> l^{\text{val}}(k, \theta, \omega)&=\left[\frac{1}{k} \sum_{j=1}^{M}\left[L_{cls}\left(g_{\omega}\left(f_{\theta}\left(x_{j}\right)\right), y_{j}\right)-\lambda^{\text{val}}(k, \theta, \omega)\right]\_{+}+\lambda^{\text{val}}(k, \theta, \omega)\right]\\\\
> &\geq \left[\frac{1}{2k} \sum_{j \in K}\left|g_{\omega}\left(f_{\theta}\left(x_{j}\right)\right)-y_{j}\right|\right]\\\\
> & \geq \frac{1}{4k} \left[\sum_{j \in K, y_{j}=0}\mathbb{1}\left(g_{\omega}\left(f_{\theta}\left(x_{j}\right)\right)\neq y_{j}\right) + \sum_{j \in K, y_{j}=1}\mathbb{1}\left(g_{\omega}\left(f_{\theta}\left(x_{j}\right)\right)\neq y_{j}\right)\right]\\\\
> &\geq \frac{1}{4}\left[\frac{1}{\alpha_0} FPR_{a=0} + \frac{1}{\alpha_1} FPR_{a=1} + \frac{1}{\beta_0} FNR_{a=0} + \frac{1}{\beta_1}  FNR_{a=1} \right]\\\\
> &\geq \frac{1}{4}\left[\min(\frac{1}{\alpha_0},\frac{1}{\alpha_1})|FPR_{a=0}-FPR_{a=1}| + \min(\frac{1}{\beta_0} ,\frac{1}{\beta_1})|FPR_{a=0}-FPR_{a=1}|\right],
> \end{aligned}
> \end{equation}
>
> where the first inequality is due to the fact that the binary cross-entropy loss is lower-bounded by its corresponding mean absolute error [3], $K$ is the set of indices of samples with top-$k$ validation loss, $\alpha_i = k\sum_{j \in K} \mathbb{1}(y_j=0,A_j=i)$ and $\beta_i = k{\sum_{j \in K} \mathbb{1}(y_j=1,A_j=i)}$. Besides, we show from experiments that our method also improves disparate impact on several datasets with different sensitive attributes.
>
> * **Explanation of main idea of contrastive learning in Section 2:** The main idea of contrastive learning is to learn a compact representation of input features on hypersphere such that samples of same labels are as close to each other as possible, while samples of different labels are as far apart as possible. In this way, the trained contrastive encoder can be readily applied to downstream classification tasks by only fine-tuning the linear layer on a small labelled set.
>
> **[Reference]**
>
> [1] Hashimoto, T., Srivastava, M., Namkoong, H., \& Liang, P. Fairness without demographics in repeated loss minimization. In ICML 2018.
>
> [2] Lahoti, P., Beutel, A., Chen, J., Lee, K., Prost, F., Thain, N., ... \& Chi, E. Fairness without demographics through adversarially reweighted learning. In NeurIPS 2020.
>
> [3] Feng, L., Shu, S., Lin, Z., Lv, F., Li, L., \& An, B. Can cross entropy loss be robust to label noise?. In IJCAI 2021.

---

### Official Review · Reviewer_s7Zh · 2022-07-11

**Rating:** 5
**Confidence:** 3
**Soundness:** 2 fair
**Presentation:** 2 fair
**Contribution:** 2 fair

**Summary:**

This paper proposed a fair contrastive learning method with gradient-based reweighing to learn a fair representation for downstream classification tasks without accessing sensitive attributes. The top-$k$ loss is employed as a surrogate fairness constraint. The convergence of the proposed method is theoretically proved and the effectiveness of the proposed method is verified on benchmark datasets.

**Questions:**

1. The motivation for this work is not strong. The paper does not do a very good job of motivating why sensitive information is infeasible in real-world scenarios. I think the paper would be more compelling if the authors could provide more concrete scenarios where sensitive information is infeasible.

2. The authors claimed that conventional methods on fairness would fail to work when there are more than one sensitive attributes. However, I think the sensitive attributes in previous works are not restricted to being single.

3. In line 119, what are the superscripts c and s?

4. In line 158, what is the subscript $+$? How does $\ell(k,\theta)$ calculate the top-$k$ average loss?

5. It is not clear to me how the proposed method learns fair representations. Can the authors comment on how the weights help to learn fair representations in contrastive learning and how the minimization of classification loss on validation samples helps to learn the proper weights? Is there any physical meaning to the weights? Do they have ground-truth values?

6. Some improvements of the proposed method are marginal. It is better to do a significance test.

7. The size of the validation set seems important. It is better to do an ablation study for it.

======================= After Rebuttal =========================

I increased my score to borderline accept.

**Limitations:**

The authors have not adequately addressed the limitations and potential negative social impact of their work.

**Strengths And Weaknesses:**

Originality: The proposed method is a combination of Min-Max fairness, importance reweighting, contrastive representation learning, and inner-product-based weight approximation.

Quality: This work is technically sound. The claims are well supported.

Clarity: Overall this paper is well written and organized. However, there are some nations being used without clearly defined.

Significance: The proposed method is insightful and provides a new perspective on learning fair representations in an unsupervised manner.

---

> ### Author Response · Authors · 2022-08-02
> **Response to reviewer s7Zh**
>
> Thanks for the comment. We'll fix the typos and include the suggested reference in final version.
>
> * **Motivation for this work:** In many real-world scenarios, it is infeasible to collect sensitive information due to legal restrictions. For example, General Data Protection Regulation (GDPR) imposes heightened prerequisites to collect and use protected features [Article 6, GDPR]. Consumer Financial Protection Bureau (CFBP) prohibits creditors from collecting or requesting information of applicants' race, color, religion, national origin, or sex, while requesting creditors to fulfill fairness [CFBP Consumer Law and Regulations, 12 CFR §1002.5].
>
> * **Fairness with multiple sensitive attributes:** Sorry for the confusion. We clarify that although some methods might be applicable under multiple sensitive attributes, many conventional fairness methods are formulated with one predefined sensitive attribute and one predefined fairness metric [1,2,3], and generalizing these methods to multiple-sensitive-attributes scenarios requires extra work and sometimes re-formulation of the whole method. We'll rewrite the sentence and make it clear in final version.
>
> * **Notion clarification:** In line 119, $c$ is the number of classes, and $s$ is the number of sensitive attributes. In line 158, the subscript $+$ means calculating the absolute value, i.e., $[x]_+ = \max(x,0), \forall x \in \mathbb{R}$. To calculate top-$k$ average loss we can simply set $\lambda(k, \theta)$ as the value of the $k$-th largest loss.
>
> * **How the proposed method learns fair representations:** Our objective is to minimize the top-$k$ average validation loss to improve fairness. Similar to DRO [4] and ARL [5], our assumption is that disadvantaged group(s) are likely to contain larger portion of hard samples, and by focusing on hard validation samples (i.e., validation samples prone to be wrongly classified), we are able to reduce the disparities between different sensitive groups. As we only have access to label information on validation set, our idea is to reweight unlabelled training samples based on gradient similarity, such that samples whose gradient directions are similar to that of top-$k$ average validation loss are upweighted. In this way, we are able to ensure the minimization of weighted contrastive loss also has a positive contribution to the decrease of top-$k$ average validation loss, thus improve fairness.
>
> * **Physical meaning of weight:**  The weight for the $i$-th training sample can be related to the angle between the gradient w.r.t the contrastive loss and the gradient w.r.t. the top-$k$ average validation loss. Greater weight indicates smaller angle between the two gradients, i.e., the training sample has a positive impact on decreasing top-$k$ average validation loss.
>
>
> * **Ground-truth values of weight:** There is no ground-truth of weight, as directly solving Eq. (3) can be very expensive and time-consuming. Instead, we consider an alternative weight according to gradient similarity.
>
> * **Significance test:** We perform t-test to validate the improvement in equalized odds by our method in Table 1-6 of main paper. The baseline is chosen as contrastive learning baseline. From the results shown in the table below, most $p$-values are less than 0.1, which validates the improvement by our method.
>
> Dataset|Sensitive Attribute|$p$-value
> -|-|-
> COMPAS|sex|0.04
> COMPAS|race|0.03
> Adult|gender|0.05
> Adult|race|0.08
> CelebA|gender|0.07
> CelebA|age|0.13
>
> * **Ablation study:** We show the effect of validation size on accuracy and equalized odds in [Fig. 1 and Fig. 2](https://docs.google.com/document/d/1Cumrg2DcYGTPR_4gpEMKrUzJW35EGvUlOni7jRubNx4/edit?usp=sharing). As shown in the figures, when the validation size is larger than $10\%$ of training size, the model's performance becomes stable in terms of accuracy and fairness.
>
> * **Limitations and and potential negative social impact:** One limitation of our work is that, although our method is conducted without sensitive information during training, we still rely on sensitive information to quantify disparities during testing since we use the metrics disparate impact and equalized odds. It remains a challenge to test fairness improvement without sensitive information during testing.
>
> **[Reference]**
>
> [1] Hardt, M., Price, E., \& Srebro, N. Equality of opportunity in supervised learning. In NeurIPS 2016.
>
> [2] Menon, A. K., \& Williamson, R. C. The cost of fairness in binary classification. In ACM FAccT 2018.
>
> [3] Krasanakis, E., Spyromitros-Xioufis, E., Papadopoulos, S., \& Kompatsiaris, Y. Adaptive sensitive reweighting to mitigate bias in fairness-aware classification. In WWW 2018.
>
> [4] Hashimoto, T., Srivastava, M., Namkoong, H., \& Liang, P. Fairness without demographics in repeated loss minimization. In ICML 2018.
>
> [5] Lahoti, P., Beutel, A., Chen, J., Lee, K., Prost, F., Thain, N., ... \& Chi, E. Fairness without demographics through adversarially reweighted learning. In NeurIPS 2020.

---

> > ### Comment · Reviewer_s7Zh · 2022-08-09
> > **Thanks for the response**
> >
> > I have read the response from the authors and appreciate their further clarifications. The authors have addressed most of my concerns. Therefore, I increased my score to 5.

---

> > > ### Author Response · Authors · 2022-08-09
> > > **Thank you**
> > >
> > > Thanks for your valuable feedback! We sincerely appreciate your time and effort in reviewing our paper.

---

### Author Response · Authors · 2022-08-05
**We are happy to answer more questions if there still exist concerns for our paper**

Dear Reviewers,

Thanks for your time and efforts in reviewing our paper. We appreciate your constructive comments. Hopefully, our response can address your concerns.

If you have further questions or confusion, we would be very happy to clarify. Thank you very much.

Best,
Authors

---

### Meta-Review · Area_Chair_r59j · 2022-08-22

**Recommendation:** Accept
**Confidence:** Certain

**Metareview:**

The paper proposes a contrastive learning approach to learn fair representations, without having explicit access to sensitive attributes. The approach leverages a small validation set with sensitive attributes to help guide training, by assigning per-sample weights based on the effect on the validation loss. All reviewers were supportive of acceptance, with a common appreciation for the novelty of the problem setting, and the modularity of the proposed approach.

The authors can consider adding citations to works operating in the related setting where there are _noisy_ protected attributes, e.g., Gupta et al., "Proxy Fairness"; Lamy et al., "Noise-tolerant fair classification"; Wang et al., "Robust Optimization for Fairness with Noisy Protected Groups".

**Award:**

No

---

### Decision · Program_Chairs · 2022-09-14

Accept